# Elimination of Eight Viruses and Two Viroids from Preclonal Candidates of Six Grapevine Varieties (*Vitis vinifera* L.) through In Vivo Thermotherapy and In Vitro Meristem Tip Micrografting

**DOI:** 10.3390/plants11081064

**Published:** 2022-04-13

**Authors:** Vanja Miljanić, Denis Rusjan, Andreja Škvarč, Philippe Chatelet, Nataša Štajner

**Affiliations:** 1Department of Agronomy, Biotechnical Faculty, University of Ljubljana, 1000 Ljubljana, Slovenia; vanja.miljanic84@gmail.com (V.M.); denis.rusjan@bf.uni-lj.si (D.R.); 2Chamber of Agriculture and Forestry of Slovenia, Agriculture and Forestry Institute Nova Gorica, 5000 Nova Gorica, Slovenia; andreja.skvarc@go.kgzs.si; 3UMR AGAP Institut, Univ. Montpellier, CIRAD, INRAE, Institut Agro, F-34398 Montpellier, France; philippe.chatelet@inrae.fr

**Keywords:** *Vitis vinifera* L., grapevine viruses and viroids, thermotherapy, micrografting

## Abstract

Viruses and virus-like organisms are a major problem in viticulture worldwide. They cannot be controlled by standard plant protection measures, and once infected, plants remain infected throughout their life; therefore, the propagation of healthy vegetative material is crucial. In vivo thermotherapy at 36–38 °C for at least six weeks, followed by meristem tip micrografting (0.1–0.2 mm) onto in vitro-growing seedling rootstocks of Vialla (*Vitis labrusca* × *Vitis riparia*), was successfully used to eliminate eight viruses (grapevine rupestris stem pitting-associated virus (GRSPaV), grapevine Pinot gris virus (GPGV), grapevine fanleaf virus (GFLV), grapevine leafroll-associated virus 3 (GLRaV-3), grapevine fleck virus (GFkV), grapevine rupestris vein feathering virus (GRVFV), grapevine Syrah virus-1 (GSyV-1), and raspberry bushy dwarf virus (RBDV)), as well as two viroids (hop stunt viroid (HSVd) and grapevine yellow speckle viroid 1 (GYSVd-1)) from preclonal candidates of six grapevine varieties (*Vitis vinifera* L.). A half-strength MS medium including vitamins supplemented with 30 g/L of sucrose and solidified with 8 g/L of agar, without plant growth regulators, was used for the growth and root development of micrografts and the subsequently micropropagated plants; no callus formation, hyperhydricity, or necrosis of shoot tips was observed. Although the overall regeneration was low (higher in white than in red varieties), a 100% elimination was achieved for all eight viruses, whereas the elimination level for viroids was lower, reaching only 39.2% of HSVd-free and 42.6% GYSVd-1-free vines. To the best of our knowledge, this is the first report of GPGV, GRVFV, GSyV-1, HSVd, and GYSVd-1 elimination through combining in vivo thermotherapy and in vitro meristem tip micrografting, and the first report of RBDV elimination from grapevines. The virus-free vines were successfully acclimatized in rockwool plugs and then transferred to soil.

## 1. Introduction

Viruses and virus-like organisms can cause severe economic problems in vitiviniculture due to (i) yield reduction, (ii) alterations of grape and wine chemical and sensorial quality, (iii) various developmental and morphological malformations in vine organs, and (iv) shortened vine life [1,2,3,4]. They cannot be controlled by conventional plant protection measures; thus, planting healthy vegetative propagation material is crucial for sustainable vine cultivation. The presence of viruses and viroids is particularly common in local, domesticated, and indigenous grapevine varieties. These are usually grown in limited areas and, due to the high concentration of the same variety cultivated on the same vineyard lands for decades, or even centuries, vine infections are usually increased, causing additional problems for clonal selection in these varieties. This is one of the main reasons why clonal selection is very difficult. Various measures for producing virus- and viroid-free grapevine material have been used, including thermotherapy [5,6], meristem tissue culture [7,8,9,10], micrografting, chemotherapy [11,12,13,14], cryotherapy [15], somaclonal embryogenesis [10,16,17,18,19], electrotherapy [20], and various combinations of these treatments [21,22,23,24]. Thermotherapy in combination with a meristem tissue culture is a widely used method. Thermotherapy is a treatment in which plants are exposed to high temperatures for a specific period of time [25]. High temperatures can inhibit virus replication or cause virus RNA degradation [26,27]. Furthermore, thermotherapy is associated with an antiviral immune defense mechanism, termed RNA silencing [27,28,29,30,31,32,33,34]. Elevated temperatures induce virus-derived small interfering RNA (vsiRNA) biogenesis and inhibit viral RNA accumulation, whereas key genes in the RNA silencing pathway were up-regulated in pear shoot meristem tips infected with apple stem grooving virus (ASGV) [33], and in pepper plants infected with tobacco mosaic virus pathotype P0 (TMV-P0) [34]. It was found that miRNAs were differentially expressed at high temperatures and miRNA-mediated target genes related to disease defense and hormone signal transduction were up-regulated in pear shoot meristem tips infected with ASGV, leading to a reduction in viral titer [32]. To increase virus elimination efficiency, thermotherapy is often combined with a meristem culture. Meristem tips consist of the apical dome and a limited number of leaf primordia, and they exclude differentiated vascular tissue [35]. The main advantages of the use of meristems are the ability to exclude pathogens present in mother plants and genetic stability [35]. The regeneration of woody plants directly from meristems is difficult; another technique that can speed up this process is micrografting. Micrografting corresponds to the placement of a meristem or shoot tip explant onto a decapitated rootstock grown under in vitro conditions [36,37]. Thermotherapy combined with meristem/shoot tip micrografting has been used to eliminate citrus viruses [38,39] and major grapevine viruses [22]. The meristem size plays an important role in the efficiency of viral entity elimination, because smaller meristems have lower survival rates but the highest virus elimination efficiency. Sanitation success also depends on the grapevine variety, virus/viroid species, their localization and interaction with plants, and treatment conditions.

Slovenia is a traditional wine-growing country, with 15,075 hectares of vineyards in 2021 (Database of Ministry of Agriculture, Forestry and Food). The Primorska wine-growing region represents 40.6% of the total Slovenian vineyard area, where a successful program of clonal selection, especially of indigenous, domesticated, and local grapevine varieties, has been taking place for decades. According to the Official Gazette of the RS N°93/05 and 101/20, all propagated vine material must undergo mandatory testing on: arabis mosaic virus (ArMV), grapevine fanleaf virus (GFLV), raspberry ringspot virus (RpRSV), tomato black ring virus (TBRV), grapevine virus A (GVA), grapevine virus B (GVB), grapevine rupestris stem pitting-associated virus (GRSPaV), grapevine leafroll-associated virus 1 and 3 (GLRaV-1, -3), and grapevine fleck virus (GFkV) (only for rootstocks). Testing on grapevine leafroll-associated virus 2 and 4–9 (GLRaV-2, -4–-9) is not obligatory but just recommended. In our previous study [40], preclonal candidates were screened for viruses and viroids using high-throughput sequencing (HTS) technology, and nine viruses and two viroids were detected. In addition to viruses with obligatory testing—GRSPaV, GFLV, GLRaV-3, and GFkV (for rootstocks)—we detected grapevine Pinot gris virus (GPGV), raspberry bushy dwarf virus (RBDV), and three grapevine fleck-similar viruses: grapevine red globe virus (GRGV), grapevine rupestris vein feathering virus (GRVFV), and grapevine Syrah virus-1 (GSyV-1). Two viroids, hop stunt viroid (HSVd) and grapevine yellow speckle viroid 1 (GYSVd-1) were also detected.

Thus, in the present study, we report on the efficiency of in vivo thermotherapy followed by in vitro meristem tip micrografting in the elimination of the eight above-listed viruses (except GRGV) and two viroids from preclonal candidates of six grapevine varieties.

## 2. Results

### 2.1. Plant Regeneration

A total of 598 meristems were isolated and micrografted, from which 51 plants were regenerated (Table 1; Figure 1a,b). To increase their number, the regenerated plants were micropropagated several times, during which period callus formation, hyperhydricity, or necrosis were never observed (Figure 1c).

Higher regeneration rates were observed in white varieties compared with red varieties (Table 1; Figure 2). Only one sample of the white variety, ‘Laški rizling’ (3/45B), infected with eight viral entities and with at least three genetic variants of GRSPaV, did not regenerate. Among the reds, one sample of ‘Pokalca’ (9/2G) and two samples of ‘Refošk’ (12/3P and 12/6P) did not regenerate (Table 1). ‘Rebula’ had the highest regeneration rate (16.7%), followed by ‘Laški rizling’ and ‘Zeleni Sauvignon’ (10.7%). Although ‘Rebula’ had the highest regeneration rate, ‘Zeleni Sauvignon’ regenerated and grew much faster during micropropagation. However, ‘Pokalca’ had the lowest regeneration rate (3.3%) (Figure 2).

### 2.2. Virus and Viroid Elimination and Vine Acclimatization

The efficiency of the elimination of viruses and viroids from regenerated plants grown in vitro for seven months was analyzed using RT-PCR. A 100% elimination rate was achieved for all viruses. However, the elimination of viroids HSVd and GYSVd-1 was significantly lower at 39.2% and 42.6%, respectively (Table 2; Appendix A). Specific RT-PCR products of the positive controls were obtained in all cases, whereas no amplicons were generated in the negative controls (Appendix A). Virus-free plants grown in vitro were successfully acclimatized in rockwool plugs, which proved to be excellent for growth and root development (Figure 3). Plants were kept in a mini greenhouse in the growth chamber (Figure 3), and then transplanted into pots (Figure 4). All the virus-free preclonal candidates will be retested after approximately three years before being officially established as certified clones.

## 3. Discussion

In this study, 28 preclonal candidates from 6 grapevine varieties *(Vitis vinifera* L.), infected with various viruses and viroids, were included into the elimination process by in vivo thermotherapy followed by in vitro meristem tip micrografting.

According to Křižan et al. [5], in vivo thermotherapy is more advisable than in vitro thermotherapy because it is less labor-intensive and provides more apical segments, whereas the shorter duration of in vitro cultivation reduces the risk of somaclonal variability. The regeneration of grapevines directly from the meristem is often difficult [41]. To improve and accelerate this process, the micrografting technique was used.

The elimination of phloem-limited viruses through the meristem tip culture is particularly effective, whereas thermotherapy, which hampers virus replication and promotes virus RNA degradation [26,27], is desirable for the elimination of other viruses [42]. In order to increase the elimination efficiency, these methods should be combined, especially in cases of mixed infections.

GLRaV-3, the main causal agent of one of the most severe grapevine diseases, grapevine leafroll disease (GLD), is phloem-limited [3]. This virus was the least prevalent in our preclonal candidate set and only one preclonal candidate (Refošk 11/4P) was found infected. Twenty-eight meristems were isolated and micrografted, and only two (7.1%) regenerated and were found to be free of GLRaV-3. The complete eradication of GLRaV-3 was achieved in several studies by thermotherapy combined with shoot apices micrografting [22], somatic embryogenesis [16,19], and cryotherapy [15]. Different efforts for GLRaV-3 elimination had thermotherapy [6,11,23] and different chemotherapeutics in combination with or not with thermotherapy [11,23]. GFkV is the causative agent of fleck disease and is also phloem-limited [43]. In Slovenia, testing for GFkV is only obligatory for rootstocks. Panattoni and Triolo [6] reported that thermotherapy had no impact on the elimination of this virus in the rootstock Kober 5BB. Bota et al. [44] reported that the combination of either a high temperature during summer in the field, or thermotherapy in the growth chamber with shoot tip culture (1–3 mm), resulted in 25% and 20% GFkV-free plants, respectively, in the ‘Manto Negro’ variety. GFkV elimination efficiency for meristem tip culture from dormant buds (0.3 mm) was 100%, whereas larger meristems (0.8 mm) resulted in a 50% lower elimination rate [9]. When thermotherapy was combined with shoot apices micrografting, complete elimination of the virus was achieved [22]. In our study, 13 GFkV-infected preclonal candidates were included into the sanitation process and 26 regenerants were obtained, which were all free of GFkV. Complete elimination was also achieved with somatic embryogenesis [10], repeated ribavirin treatment [13,45], the combination of ribavirin and oseltamivir [14], and ribavirin combined with thermotherapy [24]. Two fleck-similar viruses, GRVFV and GSyV-1, were detected for the first time in Slovenia; GRVFV was significantly more abundant [46]. GRVFV has been described to cause the mild chlorotic discoloration of leaf veins upon grafting on *V. rupestris* [43,47], whereas GSyV-1 was discovered in 2009 in an attempt to study viruses associated with decline symptoms in the ‘Syrah’ variety [48]. Although both viruses have been known for more than a decade, only one report on their elimination using a meristem tip culture and/or somatic embryogenesis has been published [10], reporting a 100% elimination, which is in accordance with our results. GFLV is the main causal agent of grapevine fanleaf disease, and is one of the most damaging grapevine viruses [1]. It is not phloem-limited and it is susceptible to heat treatment at approximately 37 °C in several rootstocks [5,6]. A meristem tip culture without thermotherapy also resulted in a high number of GFLV-free plants [8,21]. Salami et al. [21] obtained the best results when thermotherapy was combined with a meristem tip culture (0.3–0.5 mm). Thermotherapy followed by shoot apices micrografting resulted in 81% GFLV/ArMV-free plants [22]. In our study, three GFLV-infected vines of the ‘Pokalca’ variety were selected for therapy. The ‘Pokalca’ variety showed the lowest regeneration rate; only two regenerated plants were obtained. The successful elimination of GFLV had previously been achieved with somatic embryogenesis [17]. In contrast, Goussard and Wiid [49] reported that GFLV-free plants were obtained only when somatic embryogenesis was combined with thermotherapy. Chemotherapeutic agents (ribavirin and oseltamivir, independently or in a mixture) were unsuccessful in GFLV elimination from the ‘Valerien’ variety [14]. In contrast, Weiland et al. [50] reported high ribavirin efficiency (94%) in the ‘Zalema’ variety. GPGV is associated with grapevine leaf mottling and deformation disease (GLMD), which was discovered in Italy in 2012 [2]; two years later, its occurrence was reported in Slovenia [51]. It is an emerging virus in viticulture, but it has not yet been included in EU certification programs. Cytological analysis revealed the presence of GPGV particles in deep parenchyma cells [52]. Gualandri et al. [53] reported the successful sanitation of GPGV-infected vines by meristem tip culture with or without thermotherapy, whereas Turcsan et al. [10] reported that the virus elimination rates in nine vines of ‘Trilla’ and ‘Sziren’ varieties were 60% and 50%, respectively, when the meristem tissue culture was used without thermotherapy. In our previous study [40], GPGV was the most prevalent virus with 91.14% of infected vines included in the study, for which 26 samples were selected for therapy and all 49 regenerated plants were GPGV-free. Successful elimination has also been achieved through somatic embryogenesis [10] and repeated treatment with ribavirin [13]. Previous studies [27,42,54] indicated that GRSPaV and RBDV are difficult to eliminate in grapevines and raspberries, respectively, whether by thermotherapy, a meristem/shoot tip culture, or their combination, because they are presumed to infect meristematic tissues. Maliogka et al. [41] reported that successes in GRSPaV elimination by thermotherapy and shoot tip culture were significantly different in two Greek cultivars (39.62% and 92.85%), suggesting that virus elimination depends on genotype. In our study, 26 out of 28 samples were infected with this virus. Thermotherapy at 36–38 °C for at least 6 weeks and the isolation of smaller meristems, followed by micrografting to accelerate the regeneration process, resulted in the complete elimination of GRSPaV from all regenerated plants. Somatic embryogenesis was also very efficient in GRSPaV eradication [10,16,19,42], although Turcsan et al. [10] reported that when the same procedure was applied as in the other varieties with 100% virus eradication, the highest elimination rate for the ‘Sziren’ variety was 54%. Different proportions of GRSPaV-free plants were obtained with chemotherapy and its combination with other methods, such as thermotherapy and shoot tip culture [12,13,24,55]. The first report of natural infection of vines with RBDV was published in Slovenia in 2003 [56]. Outside Slovenia, there have been few reports of grapevine infections by this virus [57,58,59,60]. Although several studies have reported RBDV elimination from raspberries using different methods [27,61,62,63], to the best of our knowledge, there are no reports of RBDV elimination from grapevines. In our sample set, four preclonal candidates of the ‘Laški rizling’ variety were found to be infected with RBDV; all of them were included in the therapy process, but the candidate Laški rizling 3/45B did not regenerate at all. A total of 11 regenerated ‘Laški rizling’ vines were obtained, all of which were RBDV-free.

Although the complete elimination of all eight viruses was achieved, sanitation rates for the widely distributed viroids HSVd and GYSVd-1 were much lower in our study, i.e., 39.2% and 42.6%, respectively, compared with virus elimination. Viroids accumulate at higher titers upon experiencing a high temperature; therefore, thermotherapy alone was unsuccessful in their elimination [18]. Different elimination rates for HSVd and GYSVd-1 by meristem tissue culture have been reported [7,10]. Treatment with ribavirin was unsuccessful [45]. Somatic embryogenesis completely eliminated both viroids from four Italian varieties [18]. Turcsan et al. [10] reported that somatic embryogenesis was more efficient in eradicating HSVd than GYSVd-1.

It can be concluded that the elimination success of viruses and viroids from vines depends on several factors. The generally high elimination rates and low regeneration rates found in our study could be because smaller meristems have low survival rates but higher efficiency in virus elimination. The low regeneration rate could be also linked to the fact that meristem isolation and micrografting techniques are difficult to handle and require a high level of expertise and rapid handling to avoid explant drying and oxidation problems. Overall, it is sufficient to obtain one virus-free, regenerated plant per candidate that can be further micropropagated.

With somatic embryogenesis, high risks of somaclonal variation exist, whereas chemotherapeutics may prove highly phytotoxic. In our study, limiting the duration of the in vitro cultivation phase through in vivo thermotherapy and plant regeneration from meristems reduced the risk of genetic instability. However, regenerated plants will be carefully monitored. In addition, the elimination success rate using combined thermotherapy and meristem tip micrografting is encouraging. It remains to be seen whether our virus-free preclonal candidates will remain negative for up to about three years, when they will be retested prior to their official certification.

## 4. Materials and Methods

### 4.1. Plant Material

Eighty-two woody cuttings of preclonal candidates, without exhibiting any visual morphological symptoms regarding viruses and viroids infections, were collected of the following grapevine varieties (*Vitis vinifera* L.): two reds, ‘Refošk’ (‘Terrano’) and ‘Pokalca’ (‘Schioppettino’), and four whites, ‘Laški rizling’ (‘Welschriesling’), ‘Rebula’ (‘Ribolla Gialla’), ‘Malvazija’ (‘Malvasia d’Istria’), and ‘Zeleni Sauvignon’ (‘Sauvignon vert’). They were collected in February 2019, in the three vineyards (B, Baza; P, Pouzelce; G, Genebank) of the clonal center of Vrhpolje (STS; Vipava Valley, Primorska wine-growing region) where the clonal selection was conducted. However, preclonal candidates were selected in various vineyards in the Primorska region according to the rules on the marketing of material for the vegetative propagation of vines (Official Gazette of the RS N°93/05 and 101/20) and the OIV process for the clonal selection of vines (Resolution oiv-viti-564a-2017). Later, after vegetative propagation, they were planted in the vineyards of STS (45°50′02.2″ N 13°56′18.8″ E). One-bud cuttings were forced to bud burst and root in common tap water with no additional nutrients at room temperature (21–22 °C) at the Biotechnical Faculty, University of Ljubljana. After one month of rooting in water, well-rooted cuttings were planted into pots and transferred to the greenhouse, from which 28 plants were randomly selected for the sanitation study.

### 4.2. Virome Status of the Preclonal Candidates

Twenty-eight grapevine preclonal candidates were included in the virus/viroid elimination process (Appendix A). The viromes of preclonal candidates were investigated using the HTS of virus- and viroid-derived small RNAs and were validated with RT-PCR and Sanger sequencing [40]. All candidates harbored mixed infections. A total of 26 out of 28 candidates were found to be infected with GRSPaV and GPGV. Moreover, 13 candidates were infected with GFkV, whereas 19 and 3 candidates were infected with two fleck-similar viruses, GRVFV and GSyV-1, respectively. GLRaV-3, the least prevalent virus, was only detected in Refošk 11/4P. Three candidates representing the ‘Pokalca’ variety were infected with GFLV. RBDV was present only in candidates representing the ‘Laški rizling’ variety. All candidates were found to be infected with HSVd, whereas GYSVd-1 was also present in all candidates, except for Zeleni Sauvignon 15/3P. The highest number of viral infections per candidate was eight, in Laški rizling 3/45B, and the lowest was three, in Malvazija 23/2P (Appendix A).

### 4.3. Rootstock Source

Vialla seeds (*Vitis labrusca × Vitis riparia*) were provided by INRAE, Montpellier, France. Seeds were surface-disinfected in a laminar flow hood with a 1.66% solution of sodium dichloroisocyanurate (Sigma-Aldrich, St. Louis, MO, USA), supplemented with three drops of surfactant, Tween 20 (Duchefa Biochemie, Haarlem, The Netherlands), for 15 min with constant agitation. They were then washed three times with sterile, distilled water and plated (five seeds per plate) (Figure 5a) on a 1/4 MS basal salt medium [64], supplemented with 20 g/L of sucrose and solidified with 8 g/L of agar (all chemicals were obtained from Duchefa Biochemie). The pH was adjusted to 5.8 before autoclaving at 121 °C for 15 min. The sterile seeds were stored at 4 °C for at least two months to break dormancy. After stratification, seeds were allowed to germinate in their Petri dish in a growth chamber (LTH, Slovenia) at 25 °C, in the dark. The resulting etiolated hypocotyls (Figure 5b) were sectioned into 4–5 segments, each of which served as a rootstock (Figure 5c).

### 4.4. In Vivo Thermotherapy and In Vitro Meristem Tip Micrografting

Thermotherapy was performed in a growth chamber (Kambič, Slovenia) for a minimum of six weeks to a maximum of three months at a temperature of 36–38 °C, and a photoperiod of 16 h of light and 8 h of dark (Figure 6a). After heat treatment, the apical and axillary segments were sampled (Figure 6b) and surface-disinfected according to the following protocol. First, they were rinsed under tap water and immersed in 70% ethanol for 30 s and washed in sterile, distilled water. After the ethanol was removed, the plant material was treated in a 1.66% solution of sodium dichloroisocyanurate (Sigma-Aldrich), supplemented with three drops of the surfactant Tween 20 (Duchefa Biochemie) for 10 min with constant agitation, and then rinsed three times with sterile, distilled water. Meristem tips (0.1–0.2 mm) were aseptically excised from infected buds under 10–50X magnification using a stereomicroscope (Nikon C-LEDS, Japan) (Figure 6c). The isolated meristem tips were immediately aseptically micrografted onto the sectioned hypocotyls (Figure 5c) under a stereomicroscope and inoculated on a half-strength MS medium including vitamins [64], supplemented with 30 g/L of sucrose and 8 g/L of agar (all chemicals were obtained from Duchefa Biochemie). The pH was adjusted to 5.8 before autoclaving at 121 °C for 15 min. The micrografts were incubated at 25 °C under a light intensity of 40 μmol/m^2^/s in the growth chamber (LTH, Slovenia). The plant material obtained was micropropagated several times on a fresh medium with the same components.

### 4.5. Verification of Virus and Viroid Elimination

Virus and viroid elimination rates were determined by examining the tissues of plants maintained in vitro for seven months. Total RNA was extracted from all regenerated plants using the Monarch RNA Total Miniprep Kit (New England Biolabs). RNA concentration, quality, and purity were checked using the Agilent 2100 Bioanalyzer (Agilent Technologies, Inc., Santa Clara, CA, USA) and NanoVue Plus Spectrophotometer (GE Healthcare Life Sciences, MA, USA). cDNA was synthesized using the High-Capacity cDNA Reverse Transcription Kit (Applied Biosystems), following the manufacturer’s instructions. The PCR was performed in a 20 μL reaction volume containing 10.7 μL of nuclease-free water, 4 μL of 5× PCR buffer (Promega, Madison, WI, USA), 1.6 μL of MgCl_2_ (Kapa Biosystems, Cape Town, South Africa), 1.6 μL of dNTP mix (10 mM of each of the 4 dNTPs) (Promega), 0.5 μL of each primer, 0.1 μL of KAPA Taq DNA polymerase (Kapa Biosystems), and 1 μL of cDNA. The primers used for testing after the sanitation experiment are listed in Appendix A. Positive and negative (nuclease-free water) controls were used for each virus and viroid. Amplification was performed in a thermal cycler (Applied Biosystems, Waltham, MA, USA). Results were analyzed by electrophoresis on a 1.4% agarose gel in a 1× TBE buffer and visualized with UV light after staining with ethidium bromide.

### 4.6. Acclimatization

A few virus-free and well-developed in vitro plants per preclonal candidate were taken out of the tissue culture jars and washed with sterilized water to remove any adherent medium. They were then transferred to mini greenhouses in rockwool plugs with added perlite (Plagron) and kept in a growth chamber (Kambič, Slovenia) for two months at 25 °C with a photoperiod of 16 h of light and 8 h of darkness. During acclimatization, the plants were irrigated with MS including vitamins (Duchefa Biochemie). The vents on the mini greenhouse covers were gradually opened. The acclimatized plants were later transplanted into pots and cultivated under greenhouse conditions at the STS.

## Figures and Tables

**Figure 1 plants-11-01064-f001:**
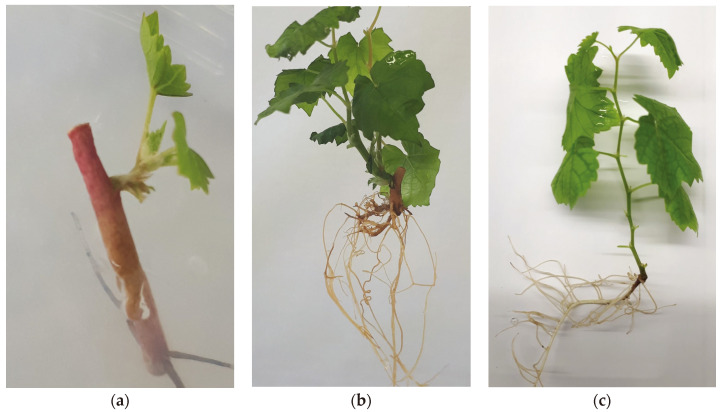
(**a**) Micrograft at the beginning of shoot and root development; (**b**) well-developed micrograft; (**c**) micropropagated grapevine separated from rootstock.

**Figure 2 plants-11-01064-f002:**
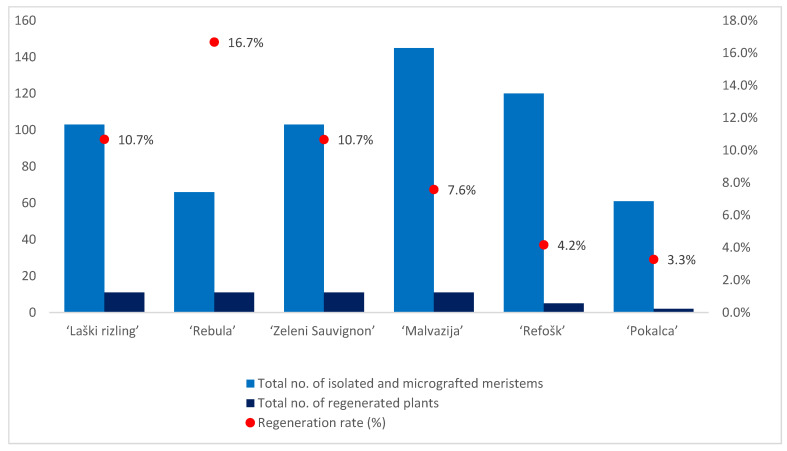
Number of isolated and micrografted meristems, number of regenerated plants, and regeneration rate per variety.

**Figure 3 plants-11-01064-f003:**
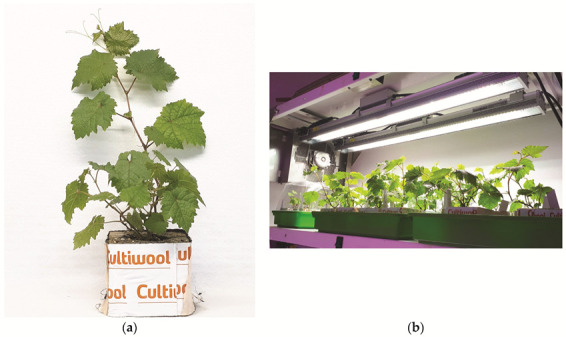
Acclimatization of virus-free plants: (**a**) in rockwool plugs; (**b**) in mini greenhouses maintained in a growth chamber.

**Figure 4 plants-11-01064-f004:**
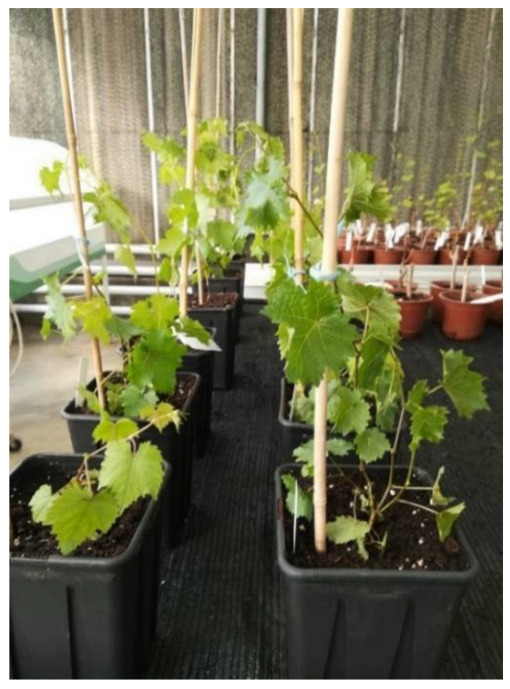
Acclimatized plants cultivated in pots in the greenhouse.

**Figure 5 plants-11-01064-f005:**
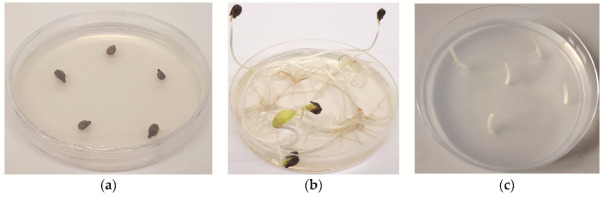
(**a**) Vialla seeds (*Vitis labrusca* × *Vitis riparia*); (**b**) etiolated hypocotyls of Vialla (*Vitis labrusca* × *Vitis riparia*); (**c**) sectioned hypocotyls into segments.

**Figure 6 plants-11-01064-f006:**
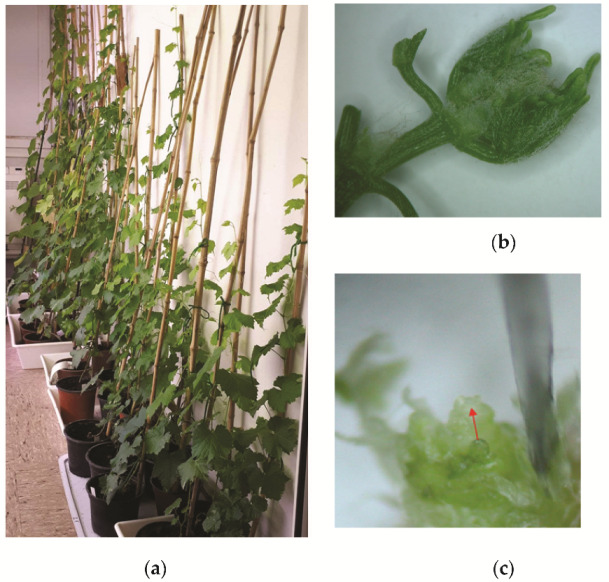
(**a**) In vivo thermotherapy; (**b**) segment prepared after in vivo thermotherapy for meristem isolation; (**c**) grapevine meristem.

**Table 1 plants-11-01064-t001:** Number of isolated and micrografted meristems, number of regenerated plants, and regeneration rate (%) per individual preclonal candidate of grapevine varieties (*Vitis vinifera* L.).

Sample Name	No. of Isolated and Micrografted Meristems	No. of Regenerated Plants	Regeneration Rate (%)
Laški rizling 3/34B	26	2	7.7
Laški rizling 3/45B	28	0	-
Laški rizling 3/64B	27	4	14.8
Laški rizling 3/56B	22	5	22.7
Rebula 15/3B	13	3	23.1
Rebula 16/1B	22	1	4.5
Rebula 19/2B	12	3	25.0
Rebula 22/3B	19	4	21.1
Zeleni Sauvignon 14/2P	18	1	5.6
Zeleni Sauvignon 14/5P	17	3	17.6
Zeleni Sauvignon 14/7P	25	2	8.0
Zeleni Sauvignon 15/2P	17	1	5.9
Zeleni Sauvignon 15/3P	26	4	15.4
Malvazija 32/1B	27	3	11.1
Malvazija 32/2B	12	1	8.3
Malvazija 32/3B	25	1	4.0
Malvazija 20/47P	13	1	7.7
Malvazija 21/8P	24	1	4.2
Malvazija 23/2P	25	1	4.0
Malvazija 23/3P	19	3	15.8
Refošk 11/4P	28	2	7.1
Refošk 12/3P	18	0	-
Refošk 12/6P	27	0	-
Refošk 12/18P	25	2	8.0
Refošk 12/19P	22	1	4.5
Pokalca 3/4P	20	1	5.0
Pokalca 3/6P	22	1	4.5
Pokalca 9/2G	19	0	-

**Table 2 plants-11-01064-t002:** Number of infected preclonal candidates before the sanitation process, number of tested vines after the sanitation process, number of virus/viroid-free vines, and elimination rate (%) per individual virus/viroid.

Virus/Viroid	No. of Infected Preclonal Candidates before the Sanitation Process	No. of Tested Vines after the Sanitation Process	No. of Virus/Viroid-Free Vines	Elimination Rate (%)
GRSPaV	26	49	49	100
GPGV	26	49	49	100
GFLV	3	2	2	100
GLRaV-3	1	2	2	100
GFkV	13	26	26	100
GRVFV	19	33	33	100
GSyV-1	3	2	2	100
RBDV	4	11	11	100
HSVd	28	51	20	39.2
GYSVd-1	27	47	20	42.6

## Data Availability

Not applicable.

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
