# Peer review of "Elimination of Eight Viruses and Two Viroids from Preclonal Candidates of Six Grapevine Varieties (Vitis vinifera L.) through In Vivo Thermotherapy and In Vitro Meristem Tip Micrografting"

_plants, 2022, doi:10.3390/plants11081064_

Round 1
Reviewer 1 Report
The manuscript by Miljanić et al. is an interesting study that describes the use of thermotherapy and meristem tip micrografting to eliminate several viruses and viroids in six grapevine varieties. Although these methods have been used for virus elimination in several other crops, this study focuses on virus problems in vitiviniculture. Overall, the manuscript is well written and provides a practical solution to viral disease in grapevines. Following are my suggestions that the authors could address-
- Please provide the full names of virus and viroid full for the first time they were used in the manuscript
- Page 2, line 52- spell out vsiRNA.
- Figure 2 has typos in percentage (for eg. 18,00 % should be 18.00 %)
- Why were there differences in the regeneration rate of grapevine varieties? Please explain in the discussion section.
- Page 7, lines 201-202 “Chemotherapeutic agents (ribavirin and oseltamivir, singly or in mixture) were unsuccessful” is an incomplete statement. Please modify.
- Figure legends of figures (1,3,4,5,6) should be expanded to give more details.
- The Discussion section needs to be modified/rewritten to remove redundancy.
- The reason for the lower elimination of viroids is not explained well. Please explain.
Author Response
The manuscript by Miljanić et al. is an interesting study that describes the use of thermotherapy and meristem tip micrografting to eliminate several viruses and viroids in six grapevine varieties. Although these methods have been used for virus elimination in several other crops, this study focuses on virus problems in vitiviniculture. Overall, the manuscript is well written and provides a practical solution to viral disease in grapevines. Following are my suggestions that the authors could address-
- Please provide the full names of virus and viroid full for the first time they were used in the manuscript
Answer: We are thankful the reviewer for this comment. Full name of each virus and viroid for the first time used in the manuscript was provided and the changes could be found in the novel version of the manuscript.
- Page 2, line 52- spell out vsiRNA.
Answer: We are thankful the reviewer for this comment. The full name was written and the changes could be found in the novel version: virus-derived small interfering RNAs (vsiRNAs).
- Figure 2 has typos in percentage (for eg. 18,00 % should be 18.00 %)
Answer: We are thankful the reviewer for this comment. The figure was corrected and the changes could be found in the modified version of the manuscript.
- Why were there differences in the regeneration rate of grapevine varieties? Please explain in the discussion section.
Answer: The regeneration rate is strongly variety dependent, as it is shown in section 2.1. Plant regeneration. In addition, it was observed that varieties ‘Refošk’ and ‘Pokalca’ that had the lowest regeneration rates from micrografted meristems ( 4.2 and 3.3.) grow more slowly also in normal conditions (in vivo), compared to other varieties included in our study. In discussion section we explained that the generally high elimination rates and low regeneration rates, found in our study, could be due to the fact that smaller meristems have low survival rates, but higher efficiency in virus elimination. The low regeneration rate could be also linked to the fact that meristem isolation and micrografting techniques are difficult to handle and require a high level of expertise and rapid handling to avoid explant drying and oxidation problems.
- Page 7, lines 201-202 “Chemotherapeutic agents (ribavirin and oseltamivir, singly or in mixture) were unsuccessful” is an incomplete statement. Please modify.
Answer: The sentence was modified: Chemotherapeutic agents (ribavirin and oseltamivir, singly or in mixture) were unsuccessful in GFLV elimination from 'Valerien' variety. Changes could be found in the novel version of manuscript.
- Figure legends of figures (1,3,4,5,6) should be expanded to give more details.
Answer: We are thankful the reviewer for this comment. Everything presented in the pictures is described in details in the manuscript.
- The Discussion section needs to be modified/rewritten to remove redundancy.
Answer: We are thankful the reviewer for this comment. In discussion section redundancy was removed.
- The reason for the lower elimination of viroids is not explained well. Please explain.
Answer: We are thankful the reviewer for this comment. Thermotherapy induces viroid replication. We added some explanation in novel version in discussion section: Viroids accumulate at higher titers upon experiencing high temperature; therefore, thermotherapy alone was unsuccessful in their elimination [18]. Different elimination rates for HSVd and GYSV-1 viroids by meristem tissue culture have been reported [7,10]. Treatment with ribavirin was unsuccessful [45]. Somatic embryogenesis completely eliminated both viroids from four Italian varieties [18]. Turcsan et al. [10] reported that somatic embryogenesis was more efficient in eradicating HSVd than GYSVd-1.
Reviewer 2 Report
The study reports about elimination of important grapevine viruses and viroids in six varieties using biotechnological tools of in vivo thermotherapy and in vitro meristem micrografting with 100% elimination success albeit, low regeneration rate in some varieties than others. The manuscript is well written with an appropriate experimental design. The results and discussion support conclusions from the study. I therefore recommend publication with minor grammatical amendments.
Author Response
The study reports about elimination of important grapevine viruses and viroids in six varieties using biotechnological tools of in vivo thermotherapy and in vitro meristem micrografting with 100% elimination success albeit, low regeneration rate in some varieties than others. The manuscript is well written with an appropriate experimental design. The results and discussion support conclusions from the study. I therefore recommend publication with minor grammatical amendments.
Answer: We are very thankful the reviewer for this comment.
Reviewer 3 Report
Overall opinion:
I think this article is well designed, good structured and it is a new approach to the methods of virus elimination in grapevine. It can be useful for all countries that are stuggling with the battle of obtaining clean plant material of precious genotypes. The research is up to date to the other research activities in this field. The procedure of micrografting is good included in the whole procedure of virus elimination. The number of viruses choosen for elimination is relevant, as well as the number of tested cultivars (also the white ones and red ones).
Comments:
100-104 Plant regeneration: the number of 598 isolated meristems compare to 51 obtained plants. Comment the numbers !
173-175 When you comment Bota et al., results you should say is 20-25% small percentage or?
264-280 when woody cuttings were taken from vineyards (month)?
337-340 I think 7 months is relevant to make virus testing? Why this was performed after 7 months?
Regards,
Zvjezdana
Author Response
I think this article is well designed, good structured and it is a new approach to the methods of virus elimination in grapevine. It can be useful for all countries that are stuggling with the battle of obtaining clean plant material of precious genotypes. The research is up to date to the other research activities in this field. The procedure of micrografting is good included in the whole procedure of virus elimination. The number of viruses choosen for elimination is relevant, as well as the number of tested cultivars (also the white ones and red ones).
Comments:
100-104 Plant regeneration: the number of 598 isolated meristems compare to 51 obtained plants. Comment the numbers !
Answer: We are thankful the reviewer for this comment. A total of 598 meristems were isolated and micrografted, from which only 51 plants were regenerated. The generally high elimination rates and low regeneration, found in our study, could be due to the fact that smaller meristems have low survival rates, but a high efficiency in virus elimination. The low regeneration rate could be also linked to the fact that meristem isolation and micrografting techniques are difficult to handle and require a high level of expertise and rapid handling to avoid explant drying and oxidation problems.
173-175 When you comment Bota et al., results you should say is 20-25% small percentage or?
Answer: We have modified the sentence and changes could be found in the novel version: Bota et al. [44] reported that combination of either high temperature during summer in the field or thermotherapy in growth chamber with shoot tip culture (1-3 mm) resulted in 25% and 20% GFkV-free plants, respectively in ‘Manto Negro’ variety.
264-280 when woody cuttings were taken from vineyards (month)?
Answer: The woody cuttings were taken in February 2019.
337-340 I think 7 months is relevant to make virus testing? Why this was performed after 7 months?
Answer: Thermotherapy was conducted in summer. Meristems were isolated in second half of July/August and at the beginning of September. During autumn and winter regenerated plants were micropropagated several times to increase their number. In March we tested all samples, and virus-free plants were acclimatized in rockwool plugs and kept in a growth chamber at our faculty in Ljubljana, and in May/June they were ready to grow in normal conditions (in soil, in greenhouse at Clonal center Vrhpolje, Vipava Valley, Primorska wine-growing region).